# Collagen Type XI Inhibits Lung Cancer-Associated Fibroblast Functions and Restrains the Integrin Binding Site Availability on Collagen Type I Matrix

**DOI:** 10.3390/ijms231911722

**Published:** 2022-10-03

**Authors:** Cédric Zeltz, Maryam Khalil, Roya Navab, Ming-Sound Tsao

**Affiliations:** 1Princess Margaret Cancer Center, University Health Network, Toronto, ON M5G 1L7, Canada; 2Laboratory Medicine and Pathobiology, University of Toronto, Toronto, ON M5S 1A8, Canada; 3Departments of Medical Biophysics, University of Toronto, Toronto, ON M5G 1L7, Canada

**Keywords:** *COL11A1*, cancer-associated fibroblast, non-small cell lung cancer, integrin, collagen

## Abstract

The tumor microenvironment, including cancer-associated fibroblast (CAF), plays an active role in non-small cell lung cancer (NSCLC) development and progression. We previously reported that collagen type XI and integrin α11, a collagen receptor, were upregulated in NSCLC; the latter promotes tumor growth and metastasis. We here explored the role of collagen type XI in NSCLC stroma. We showed that the presence of collagen type XI in collagen type I matrices inhibits CAF-mediated collagen remodeling and cell migration. This resulted in the inhibition of CAF-dependent lung-tumor cell invasion. Among the collagen receptors expressed on CAF, we determined that DDR2 and integrin α2β1, but not integrin α11β1, mediated the high-affinity binding to collagen type XI. We further demonstrated that collagen type XI restrained the integrin binding site availability on collagen type I matrices, thus limiting cell interaction with collagen type I. As a consequence, CAFs failed to activate FAK, p38 and Akt one hour after they interacted with collagen type I/XI. We concluded that collagen type XI may have a competitive negative feedback role on the binding of collagen type I to its receptors.

## 1. Introduction

Cancer-associated fibroblast (CAF) is one of the major cellular components of the tumor microenvironment that plays an active role in non-small cell lung cancer (NSCLC) tumorigenicity [1]. CAFs contribute to the deposition and remodeling of extracellular matrices, e.g., collagens, influencing tumor progression, and metastasis [2,3].

Collagen type XI is a minor fibrillar collagen composed of α1(XI), α2(XI) and α3(XI) chains. Defects in the *COL11A1* gene that codes the α1 chain leads to skeletal disorders, such as Marshall and Stickler syndromes, characterized by eye problems, hearing loss and articular abnormalities [4,5]. Collagen type XI is mostly expressed in cartilage by chondrocytes, where it co-polymerizes with collagen type II and type IX [6]. Furthermore, collagen type XI nucleates self-assembly of cartilage fibrils and controls the diameter of the fibrils [7]. Multiple studies have described a high expression of collagen type XI in many cancers, whereas there is a low or no expression in normal tissues [8,9,10], suggesting that the expression of collagen type XI can serve as a cancer biomarker [11,12]. Collagen type XI has been proposed as biomarker for activated CAFs in 13 epithelial cancer types [13], including breast, ovarian and lung cancers. *COL11A1* is often found in gene signatures associated with the transforming growth factor β (TGF-β) signaling pathway [14,15]. Collagen type XI tends to promote tumor progression, since the siRNA-knockdown of *COL11A1* in ovarian and head and neck tumor cell lines significantly reduces cell invasion and proliferation [16,17]. In addition, collagen type XI has been reported to induce chemoresistance of ovarian cancer, and an Akt inhibitor is able to restore cell sensitivity to cisplatin through the inhibition of the *COL11A1* expression [18,19]. In NSCLC, our group previously showed that *COL11A1* was upregulated in tumor stroma compared with normal lung tissue [20] and we further identified *COL11A1* as one of the differentially expressed genes upregulated in CAFs vs. normal lung fibroblasts [15]. Although it is established that *COL11A1* appears to be mostly expressed by tumor stromal cells, not much is known about the effect of collagen type XI on CAF function.

In the present study, we have explored how the presence of collagen type XI in the matrix could influence CAFs’ features. Unexpectedly, we show that collagen type XI inhibits CAF-mediated collagen remodeling and cell migration in collagen type I matrices. We determined integrin α2β1 and discoidin domain receptor 2 (DDR2) as the CAF receptors that interact with collagen type XI and mediate its effects. We further demonstrated that collagen type XI restrains the integrin binding site availability on collagen type I matrices.

## 2. Results

### 2.1. Collagen Type XI Inhibits Lung CAF Features

We first determined whether CAFs were able to interact with collagen type XI as they do with collagen type I. Lung primary CAFs, previously isolated from NSCLC resected tumors [1], were seeded on monomeric (mono) or fibrillar (fib) collagens (Figure 1A). Bovine serum albumin (BSA) and fibronectin have been used as negative and another positive control for cell adhesion, respectively. CAFs attached similarly to both collagen type I and collagen type XI, in their monomeric or fibrillar form, suggesting that collagen type XI could also serve as a scaffold for CAF migration and CAF-mediated matrix remodeling.

The ability of CAFs to remodel the fibrillar collagen matrix (in vitro recognized as a collagen gel contraction) has been shown to affect tumor invasion and growth [2]. To study how collagen type XI affects matrix remodeling, CAFs were embedded in a matrix of collagen type I or collagen type XI, or in a mixture of both collagens at different ratios, approaching in vivo conditions, where collagen type I is more abundant than collagen type XI. Surprisingly, CAF poorly remodeled the collagen type XI matrix as shown by the weak collagen type XI gel contraction (−70% compared with collagen I at 24 h, *p* = 0.0004) (Figure 1B). Furthermore, the presence of collagen type XI delayed the remodeling of collagen type I matrices and this inhibition was dependent on the collagen type XI concentration.

In addition to reorganizing stromal collagen matrices, CAFs regulate tumor cell invasion by forming tracks in the stroma matrix and leading CAF-dependent tumor cell migration [21]. We investigated the effect of collagen type XI on this CAF phenotype using a spheroid model, where CAF homospheroids or heterospheroids composed of CAF with lung tumor cells were embedded in a matrix of collagen type I or a mixture of collagen type I and collagen type XI at a 3:1 ratio (Figure 1C). We selected the H1573, H2073 and H358 lung tumor cell lines for their weak ability to invade collagen type I matrices (Appendix A). To distinguish the migration of the different cell types in the spheroid model, CAFs and tumor cells were stained in green and red, respectively. The presence of collagen type XI in collagen type I matrices inhibited CAF migration (−81%, *p* = 0.00015; Figure 1C,D), resulting in the inhibition of migratory tracks and thus the reduction of CAF-dependent tumor cell invasion (−84%, *p* = 0.00058; Figure 1C,E). To confirm that collagen type XI does not directly inhibit tumor cell migration, we used two lung tumor cell lines (H1975 and H2009) that are able to invade collagen matrices independent of CAFs. We did not see any inhibitory effect of collagen type XI on the invasion of the tested lung tumor cells (Appendix A), suggesting that collagen type XI specifically inhibited CAF migration.

### 2.2. CAFs Interact with Collagen Type XI through Integrin α2β1 and DDR2

To understand the mechanism of the collagen type XI inhibitory effect on CAFs, we investigated which collagen receptor(s), i.e., integrin(s) or discoidin domain receptors, could interact with collagen type XI. We first analyzed by RT-qPCR the expression of integrins α1 (ITGA1), α2 (ITGA2), α10 (ITGA10) and α11 (ITGA11) subunits, and discoidin domain receptors DDR1 and DDR2 in CAF (Figure 2A). CAFs express ITGA2, ITGA11 and DDR2, but lack expression of ITGA1, ITGA10 and DDR1. We then checked which CAF integrin(s) could bind to collagen type XI, using the C2C12 cell line that does not originally express any collagen-binding integrins. C2C12 cells overexpressing either ITGA1 (C2C12-α1), ITGA2 (C2C12-α2) or ITGA11 (C2C12-α11) [22] were assessed for cell adhesion on fibrillar collagen type XI (Figure 2B). Fibrillar collagen type I and fibronectin were used as positive controls in this assay, while BSA was used as the negative control. C2C12-α1 cells, which have low affinity for fibrillar collagens, did not bind to collagen type XI. C2C12-α2 showed similar binding to collagen type I and type XI, whereas C2C12-α11 bound poorly to collagen type XI, indicating that integrin α2β1 is the main collagen-binding integrin on CAFs that could interact with collagen type XI. We have confirmed the direct interaction between integrin α2β1 and collagen type XI by solid phase assay (Figure 2C). The integrin α2β1 binding to collagen type XI was specific, since no binding occurred on fibronectin. Integrin α2β1 also bound to heat-denatured collagens as previously shown for collagen type I [23]. Furthermore, integrin α2β1 interacted to collagen type XI with high affinity, with a dissociation constant K_d_ of ~4–5 nM (Figure 2D). We further analyzed the direct interaction between DDR2 and collagen type XI by solid phase assay (Figure 2E). DDR2 bound specifically to collagen type XI with no binding to heat-denatured collagen type XI or fibronectin. The DDR2-collagen type XI interaction also showed high affinity with a K_d_ of ~2–5 nM (Figure 2F).

### 2.3. Collagen Type XI Restrains Collagen Type I Availability for Cells

C2C12-α11 cells, which bind poorly to collagen type XI, also displayed a reduced collagen type I/XI matrix contraction to a similar extent to what we observed with CAFs (Appendix A), suggesting that collagen type XI may restrain cell binding to collagen type I. To assess this hypothesis, we performed a cell adhesion assay on a mix of both collagen type I and collagen type XI (3:1 ratio). Since integrin α2β1 interacted with high affinity to both collagens, we took advantage of the C2C12-α11 cells, which bind to collagen type I with high affinity but poorly to collagen type XI. On monomeric collagens, where collagens were present as single molecules, the presence of collagen type XI did not affect the binding of C2C12-α11 cells on collagen type I (Figure 3A). In contrast, on fibrillar collagen, where collagens are incorporated into fibrils to form a matrix, the attachment of C2C12-α11 cells to collagen type I was inhibited by the presence of collagen type XI, indicating that collagen type XI blocked access to the integrin site on collagen type I. Cell adhesion inhibition on fibrillar collagens was significantly dependent on the concentration of collagen type XI in the different ratios of collagen type I/XI (the total collagen concentration was constant) used to form the collagen matrices (Figure 3B). We did not observe any inhibition of cell adhesion on monomeric collagens, even with a high concentration of collagen type XI (Figure 3C). To ensure that the cell-adhesion inhibition was not due to the decrease of collagen type I concentration within the different ratios, we repeated the experiment on fibrillar collagen, keeping the concentration of collagen type I constant, and showed similar results (Appendix A). Moreover, cell adhesion assay on fibrillar collagen type I alone at the different concentration used in the different ratios did not display any inhibition (Appendix A), indicating that collagen type XI was responsible for the inhibition of C2C12-α11 cell adhesion to collagen matrices. We have measured the half-maximal inhibitory concentration (*IC_50_*) of collagen type XI to inhibit cell binding to fibrillar collagen type I (Figure 3D). We determined an *IC_50_* = 560 nM of collagen type XI.

Altogether, these results demonstrated that the presence of collagen type XI restricts the integrin α11β1 binding site on collagen type I matrices. Since both integrin α2β1 and α11β1 recognize the same sequence on collagen type I [24], we suggest that collagen type XI may also inhibit the interaction between integrin α2β1 and collagen type I.

### 2.4. Defect in FAK, p38 and Akt Activation in Presence of Collagen Type XI

We studied the mechanism of the collagen type XI inhibitory effect in CAFs by analyzing the signaling pathways of collagen receptors, integrin α2β1, integrin α11β1 and DDR2, expressed in NSCLC CAFs. We investigated the activation of the integrin-dependent focal adhesion kinase (FAK) that is known to induce cell migration [25], Akt phosphorylation, which was shown to be involved in DDR2-mediated cell migration [26,27], and the MAPK p38, JNK and ERK, shown to be part of the collagen-binding integrin signaling [25,28,29]. FAK was activated when serum-starved CAFs were seeded on a fibrillar collagen type I matrix (Figure 4A). However, in the presence of collagen type XI in the collagen type I matrices, we observed a significant defect in FAK phosphorylation at early time points (Figure 4A,B). When CAFs are seeded on collagen type XI only, FAK phosphorylation was not inhibited at an early time point but decreased by 24 h compared with CAFs on collagen type I (Appendix A). We observed a similar signaling pattern regarding the phosphorylation of p38 and Akt, where activation was significantly inhibited 1 h after CAFs were seeded on the collagen I/XI matrix (Figure 4B–D) and Akt phosphorylation was not sustained in cells seeded on collagen type XI only (Appendix A). In contrast, ERK and JNK phosphorylation was unchanged in cells seeded on collagen type I or collagen type I/XI matrices (Figure 4A,E,F). These results indicate that the presence of collagen type XI in the collagen I matrix inhibited early activation of FAK, p38 and Akt in CAFs, which could result from collagen type I limited availability for CAF integrins, as described above. This, in combination with the defect of FAK and Akt long-term activation, would explain the inhibitory effect of collagen type XI on CAF-mediated collagen remodeling and cell migration (Figure 5).

## 3. Discussion

We have demonstrated that the presence of collagen type XI in collagen type I matrices inhibited CAF-mediated migration and collagen lattice remodeling, with the consequence of inhibiting CAF-dependent lung-tumor cell invasion. We found that integrin α2β1and DDR2 are the CAF receptors that interact with collagen type XI. Moreover, we demonstrated that collagen type XI restrained the integrin α2β1 and α11β1 binding site availability on collagen type I matrices. This resulted in the inhibition of FAK, p38 and Akt activation when CAFs were interacting with fibrillar collagen type I/XI matrices.

During the past years, an important question emerged regarding the role of CAFs in tumors: CAFs as friend or foe? This CAF dilemma is essentially illustrated by studies in pancreatic ductal adenocarcinoma (PDAC), in which particular sub-populations of CAFs were depleted. The deletion of FAP+CAF slows tumor growth [30], whereas the suppression of α-smooth muscle actin (SMA)-expressing CAFs mediates immunosuppression and decreases survival [31]. Moreover, Rhim et al., show that some stromal components may inhibit rather than support tumorigenicity [32]. The discrepancy of CAF functions specifically in PDAC, has been explained by CAF heterogeneity, and different sub-classes of CAFs, such as SMA-high CAFs, have further been described as a tumor-adjacent TGF-β-driven population with different inflammatory properties from SMA-low CAFs [33]. It is interesting that *COL11A1* is among the 11-gene signature that represents TGF-β-CAF significant enriched genes in myofibroblastic CAFs [34]. Therefore, we suggest that the sub-populations of CAF expressing collagen type XI may exhibit anti-tumoral activities rather than promoting tumorigenicity.

The overexpression of collagen type XI in different cancers is associated with poor survival [9,13], and collagen type XI has been shown to promote tumor progression and chemoresistance, especially in ovarian cancer [16,19]. In contrast, we showed that collagen type XI inhibited the migration of NSCLC CAFs, which in turn inhibited CAF-dependent lung-tumor cell invasion. However, we did not investigate the effects of collagen type XI on lung tumor cells, which express DDR1 that is not present in CAFs and can therefore differently modulate the tumor-cell behavior.

We have previously shown that both *ITGA11* and *COL11A1* genes were upregulated in NSCLC CAFs compared with normal lung fibroblasts [15]. There is also a strong significant correlation between *ITGA11* and *COL11A1* expression in an NSCLC patient cohort (UHN (GSE50081), Spearman r = 0.8, *p* value < 0.0001), suggesting that integrin α11 may mediate collagen type XI effects. However, here we showed that despite their high expression in CAF, integrin α11 bound weakly to collagen type XI. Instead, both integrin α2β1 and DDR2 interacted strongly with collagen type XI. It has been shown that integrin α2β1 mediates collagen remodeling and cell migration and invasion in different cancers [35,36,37,38]. Integrin α11β1 is a major collagen receptor on CAFs that induces collagen remodeling [29]. Integrin α11β1 bound poorly to collagen type I in the presence of collagen type XI and has a weak interaction with collagen type XI, leading to reduce CAFs ability to reorganize the collagen matrix. Integrin α2β1 is also involved in collagen remodeling. Since both integrin α2β1 and α11β1 recognize the same GFOGER motif on collagen type I [24], collagen type XI would inhibit the interaction between integrin α2β1 and collagen type I, thus reducing collagen remodeling. Some studies show that the supramolecular assembly of collagen limits the availability of the integrin binding sites [39,40,41,42]. Furthermore, chondrocytes seeded on fibrils of cartilage collagens containing collagen type II and XI displayed a reduced attachment compared with adhesion on monomeric cartilage collagens [40]. The phosphorylation of FAK is the first signaling event that occurs following the binding of integrins with collagens [43], and the inhibition of FAK activation abrogates cell migration in a collagen matrix [25]. In addition, integrin α2β1 also signals through Akt [44], and a crosstalk between FAK and Akt has been shown to promote cell motility in fibroblasts [45]. We found that in CAFs seeded on collagen type I/XI matrices, but not on collagen type XI only, early FAK, p38 and Akt phosphorylation was reduced. This result suggests that collagen type XI may inhibit integrin α2β1- and α11β1-mediated FAK activation, α2β1-mediated p38 and Akt activation by restraining CAF interaction to collagen type I, which could thus explain the inhibition of cell migration. Additionally, DDR2 signaling might be inhibited by the presence of collagen type XI, since DDR2 regulates the migration of fibroblast in collagen matrices [46,47]. Similar to integrin α2β1, DDR2 could also signal through Akt [48]. Therefore, the inhibition of an integrin α2β1-DDR2 cross-talk by collagen type XI cannot be excluded, as both integrins and DDR2 have been shown to collaborate [49,50]. For instance, DDR2 influences CAF-mediated collagen remodeling by regulating integrin activation [50]. Rada et al., have described collagen type XI as a driver of cisplatin resistance in ovarian cancer through binding to DDR2 and integrin α1β1 [27]. Although we showed direct interaction between DDR2 and collagen type XI, we found that integrin α1β1, which was not expressed in our CAFs, did not bind to fibrillar collagen type XI; that is in agreement with the low affinity that integrin α1β1 has for fibrillar collagens [51,52].

Limitations of the study: We have shown that collagen type XI restrained interaction of integrin α11β1 with collagen type I, but we are lacking direct evidence in regard to integrin α2β1. We did not find an appropriate method to test this hypothesis for integrin α2β1, due to: (1) integrin α2β1 binds to both collagens; (2) this inhibition occurs only when collagens form fibrils. Instead, we speculated based on the sequence recognition by both integrins on collagen type I and we assumed that since they are known to both interact with the same collagen type I sequence, in this context if integrin α11β1-collagen type I interaction is inhibited, this should also be true for integrin α2β1. Another limitation is the use of bovine collagens instead of human collagens due to technical limitation. Although collagens are highly conserved between the two species (identity >95% between the human and bovine protein sequence of both COL1A1 and *COL11A1*), the interaction between bovine collagens and human integrins may differ from the in vivo affinities in human tumors, and small variations could thus exist.

## 4. Materials and Methods

### 4.1. Cell Lines

Primary CAFs used in the present study were isolated from NSCLC specimens, as previously reported [1]. The study was conducted in accordance with a protocol approved by the University Health Network (UHN) Research Ethics Board. NSCLC tissues were collected with informed consent from all patients. An overview of CAFs used in this study is presented in Table 1. CAF cells were cultured in Dulbecco’s Modified Eagle’s medium (DMEM) supplemented with 10% fetal bovine serum (FBS; Thermo Scientific, Burlington, ON, Canada). All CAF primary cultured cells were maintained and used at early passage (passage 2–5). Murine C2C12 myoblasts stably expressing either human integrin α1 (C2C12-α1), human integrin α2 (C2C12-α2) or human integrin α11 (C2C12-α11) were obtained from Prof. Donald Gullberg (University of Bergen, Norway) and were cultured in DMEM supplemented with 10% FBS. The NCI-H1573, H2073, H358, H1975 and H2009 NSCLC cell lines were obtained from the American Type Culture Collection (Manassas, VA, USA) and were cultured in RPMI 1640 media supplemented with 10% FBS.

### 4.2. Cell Adhesion on Collagen Matrices

For cell adhesion on monomeric collagens, plates (24-well) were coated with 500 µL of Bovine PureCol^®^ collagen type I (10 µg/mL; Advanced BioMatrix, San Diego, CA, USA), bovine collagen type XI (10 µg/mL; Chondrex, Redmond, WA, USA) or a mixture of both collagens at the final concentration of 10 µg/mL diluted in phosphate buffered saline (PBS).

For cell adhesion on fibrillar collagens, a collagen gel solution was prepared by mixing 5 parts of 2X DMEM, 1 part of 0.2 M HEPES at pH 8.0 and 4 parts of 3.1 mg/mL of collagen type I (4.1 µM), type XI (3.4 µM) or of a mix of both collagens for a final concentration of 1.24 mg/mL. The different concentrations of collagen type I and type XI in the different ratios used are indicated in the Table 2. A thin layer of the collagen mixture was coated on a 24-well plate.

Human plasma fibronectin (Sigma-Aldrich, Oakville, ON, Canada) was used as a positive control for cell attachment at the concentration of 2 µg/mL in 500 µL of PBS/well, whereas BSA (Sigma-Aldrich) coating (2%) was used in all adhesion experiments as an internal negative control. The coated plates were incubated for 1 h at 37 °C. Plates were then blocked with 2% BSA for 1 h at 37 °C. Cells were trypsinized and washed twice with DMEM without FBS, and 2 × 10^5^ cells/well were seeded on the plate and incubated for 40 min at 37 °C. Following incubation, non-adherent cells were removed by washing 3 times with PBS. Cells were fixed with 1.1% glutaraldehyde for 10 min at room temperature, washed 3 times with distilled water and stained with 0.1% crystal violet for 20 min at room temperature. Crystal violet was released from cells using 10% of acetic acid. Absorbance was read at 595 nm (Spectramax^®^ Plus 384, Molecular Devices, San Jose, CA, USA).

### 4.3. Collagen Gel Contraction Assay

This assay was performed as previously described [29]. Contraction of collagen was performed in 96-well plates. The plates were coated with 2% BSA in PBS, incubated at 37 °C overnight and then washed 3 times with sterile PBS before use. Cells were trypsinized, washed 3 times with serum free DMEM and diluted in 2X DMEM. The collagen solution was mixed on ice: 5 parts of 2X DMEM containing 6 × 10^5^ cells/mL, 1 part of 0.2 M HEPES at pH 8.0 and 4 parts of 3.1 mg/mL of collagen type I, type XI or of a mix of both collagens for a final concentration of 1.24 mg/mL. An amount of 100 µL collagen/cell suspension was added to each well. The plate was immediately incubated at 37 °C to allow gels to form. After 1 h, 100 µL DMEM was added to each well to float the gel. The plate was then incubated at 37 °C for 24 h. The contraction process was measured at 5 and 24 h by analyzing the diameters under an inverted microscope with an ocular micrometer.

### 4.4. 3D-Matrix Invasion Assay

This assay was performed as previously described [1]. Confluent cell monolayers were trypsinized and suspended in PBS (2 × 10^6^ cells/mL). The Vybrant^®^ DiO green dye and DiD red dye (1:200 dilution; Invitrogen, Mississauga, ON, Canada) was added to fibroblasts and lung tumor cell lines, respectively, and incubated for 15 min at 37 °C. Cells were then washed twice with PBS. Either tumor cells or fibroblasts were diluted to 8 × 10^5^ cells/mL in cell cultured medium and a ratio of 1:4 (tumor cells:fibroblasts) was used for making heterospheroids. A drop of 40 µL of cells was added in each well of 96-well Sphera-plate (VWR, Mississauga, ON, Canada) to form homo- and heterospheroids. Cells began to form spheroids after 1 day. On day two, the spheroids were flushed in collagen gel and were visualized under a Zeiss LSM700 confocal fluorescent microscope (Zeiss, Toronto, ON, Canada) using 5 × 0.25 NA objective at day 2 post-embedding. The 3D-matrix invasion areas were analysed using texture analysis available as a plugin for the freeware Image J analysis software (http://imagej.nih.gov/ij/index.html) to evaluate the pixels in different directions around the spheroid, which is a measure of the invading area.

### 4.5. Reverse-Transcriptase/Quantitative PCR (RT-qPCR) Expression Profiling

This assay was performed as previously described [3]. Total RNA was isolated from cultured cells using the Qiagen RNEasy Kit (Qiagen, Venlo, The Netherlands). Total RNA was reverse-transcribed to synthesize 1 µg of cDNA, using Superscript III^®^ (Life Technologies, Burlington, ON, Canada). The cDNA was diluted and 10 ng was used for each quantitative PCR reaction performed by CFX96 Touch™ RT-PCR Detection System (BioRad, Hercules, CA, USA), using 2X SYBR Green master mix (Life Technologies). The ΔCT value for each gene was log2 transformed and normalized using the housekeeping gene RPS13. For the primer list, please refer to Table 3.

### 4.6. Solid Phase Assay

Collagen type I (50 µg/mL), collagen type XI (50 µg/mL) and fibronectin (10 µg/mL) diluted in PBS were immobilized on a 96-well plate (50 µL/well) overnight at room temperature. To obtain heat-denatured collagens, collagen type I and type XI were heated at 50 °C for 30 min prior to immobilization. Wells were then blocked with 150 µL of 0.05 mg/mL κ-casein in 0.05% tween 20/PBS (PBS-T) (DDR2 binding) or 1% BSA in PBS-T (integrin α2β1 binding) for 1 h. The plate was then washed once with the blocking buffer. Recombinant human DDR2-Fc chimera protein (Sino Biological, Wayne, PA, USA) and recombinant human integrin α2β1 (R&D Systems, Minneapolis, MN, USA) were diluted in the incubation buffer (same as blocking buffer for DDR2, for integrin α2β1, the blocking buffer was supplemented with 2 mM of MgCl_2_) and 50 µL of the solution was added to each well of the plate for 3 h at room temperature under gentle agitation. The plate was then washed six times with incubation buffers and 50 µL of goat anti-human IgG (Fc specific) linked to horseradish peroxidase (HRP) (1:4000 dilution, for DDR2 detection) or mouse anti-human integrin α2β1 (1:1000 dilution) for 1 h at room temperature under gentle agitation. After six washes with incubation buffer, 50 µL of anti-mouse HRP (1:5000 dilution) was added for integrin α2β1 detection and incubated for 1 h at room temperature under gentle agitation. After six more washes, 100 µL of 3,3’,5,5’-tetramethylbenzidine (TMB) and peroxide solution (mixed at equal volume; Thermo Scientific) was added in each well and incubated 15 min at room temperature in the dark. Absorbance was read at 400 nm. Estimates for the dissociation constant K_d_ were obtained using the Scatchard plot:$$\frac{\left[PL\right]}{\left[L\right]}=f\left(\left[PL\right]\right)$$
where [*L*] is the free ligand concentration (DDR2 or integrin α2β1) and [*PL*] is the concentration of bound ligand (determined by absorbance), $\frac{-1}{Kd}$ being the slope of the line.

### 4.7. Western Blot Analysis

This was performed as previously described [1]. Fibroblast primary cultured cells were homogenized in lysis buffer (RIPA buffer (Sigma-Aldrich) including protease (Roche Diagnostics Canada, Mississauga, ON, Canada) and phosphatase (Thermo Scientific) inhibitors), and the lysates were cleared by centrifugation. Protein samples were fractionated on SDS/polyacrylamide gels and transferred to polyvinylidene fluoride membranes. The membranes were blocked with 5% nonfat dry milk and incubated with primary antibodies overnight. Blots were incubated with rabbit anti-human phospho-FAK (Tyr397), rabbit anti-human phospho-p38 MAPK (Thr180/Tyr182), rabbit mAb anti-human phospho-p44/42 MAPK (ERK1/2) (D13.14.4E), rabbit mAb anti-human phospho-Akt (Ser473) (D9E) (1:1000 dilution (1:2000 for pERK); Cell Signaling Technology, Danvers, MA, USA) or mouse mAb anti-human phospho-JNK (Thr183/Tyr185) (G-7) (1:1000 dilution; Santa Cruz Biotechnology, Dallas, TX, USA) overnight at 4 °C. After incubations with the appropriate secondary antibodies, immunoreactive protein bands were detected by enhanced chemiluminescence (Roche Diagnostics). Equal protein loading was confirmed by reprobing the blot with antibodies against total FAK, p38, Akt and ERK (1:1000 dilution (1:5000 for ERK); Cell Signaling Technology, MA, USA), JNK (1:1000 dilution, Santa Cruz) or β-actin (1:5000 dilution; Millipore Sigma, Oakville, ON, Canada). Band intensities in Western blots were quantified using Image J software.

### 4.8. Measurement of IC_50_ Value

The half-maximal inhibitory concentration (*IC_50_*) of collagen type XI to inhibit cell binding to fibrillar collagen type I has been calculated from the plot: $\%\;of\;adhesion=f(log_{10}\left[CollXI\right]$) with a trend that has the function $y=aln\left(IC_{50}\right)+b$. *IC_50_* was then determined from the equation:$$log_{10}\left(IC_{50}\right)=e^{\left(\frac{(\left(Max-\left(\frac{Max-Min}{2}\right)\right)-b}{a}\right)}$$
where *Max* and *Min* are the maximum and minimum values of percentage of adhesion, respectively.

### 4.9. Statistical Analysis

Results are expressed as the mean ± standard deviation of at least three replicates and are representative of three independent experiments. Statistical significance was assessed using unpaired Student’s t-tests or a Mann–Whitney test when replicates were more than 4, with *p* < 0.05 being considered significant. Calculations were performed using R (4.1.1).

## 5. Conclusions

Our findings demonstrate that collagen type XI serves as a negative feedback mechanism for a collagen-mediated NSCLC CAF function in lung cancer tumor growth. Therefore, the traditional view of the CAFs as pro-tumorigenic cells needs reconsideration, as certain CAF subtypes with high expression and affinity binding to collagen type XI might have antitumorigenic features.

## Figures and Tables

**Figure 1 ijms-23-11722-f001:**
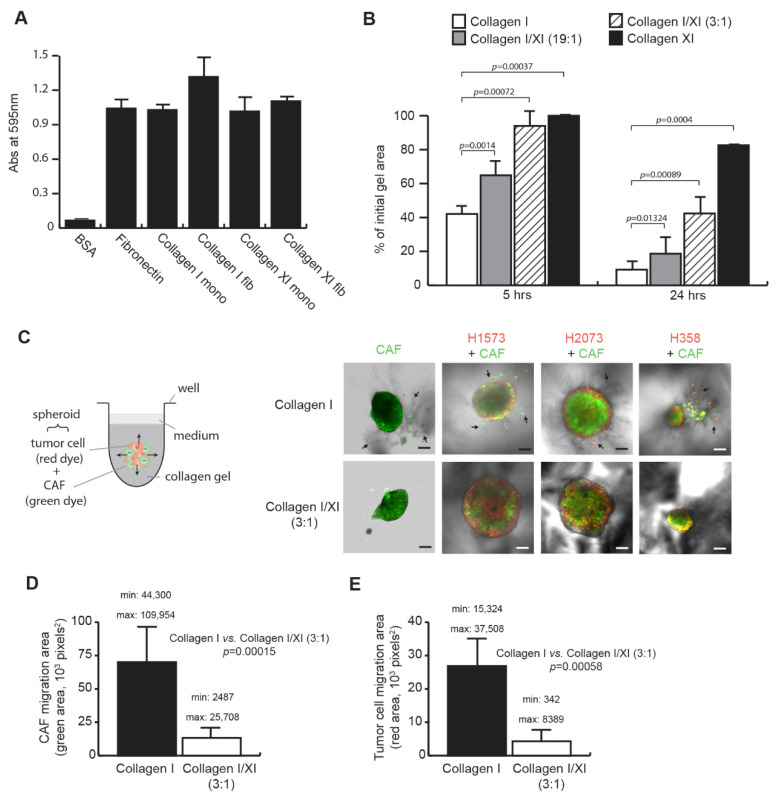
Collagen type XI inhibits cancer-associated fibroblast (CAF) functions. (**A**) CAFs bind to collagen type XI. CAFs were seeded on monomeric (mono) or fibrillar (fib) collagen type I and collagen type XI for 40 min. Cells were stained with crystal violet and absorbance was read at 595 nm. Bovine serum albumin (BSA) and fibronectin were used as negative and positive control for cell adhesion, respectively, (*n* = 4, mean ± SD). (**B**) Collagen type XI delays CAF-mediated collagen matrix remodeling. CAFs were embedded in a matrix of collagen type I, collagen type XI or a mix of both collagens at a ratio of 19:1 and 3:1 (collagen type I:collagen type XI), and allowed to contract the matrix for 24 h. Area of the collagen matrices has been measured at 5 and 24 h. Statistics were performed using a Mann–Whitney test (*n* = 8, mean ± SD). (**C**) Collagen type XI inhibits CAF-mediated lung tumor cell migration. Heterospheroids of CAFs (stained with a green dye) and lung tumor cells (stained with a red dye) were embedded in a matrix of collagen type I or a mix of collagen type I and collagen type XI at a 3:1 ratio. The invasion into the collagen matrices was visualized for 2 days using confocal microscopy, scale bars: 200 µm. Arrows denote tracks in the collagen matrix made by migrating CAFs. The migrated area in the collagen matrices was quantified for CAF (**D**) and lung tumor cells (**E**) for the three different heterospheroids presented in **C** and has been averaged. Minimum and maximum values are indicated in each graph. Statistics were performed using a Mann–Whitney test (*n* = 8, mean ± SD).

**Figure 2 ijms-23-11722-f002:**
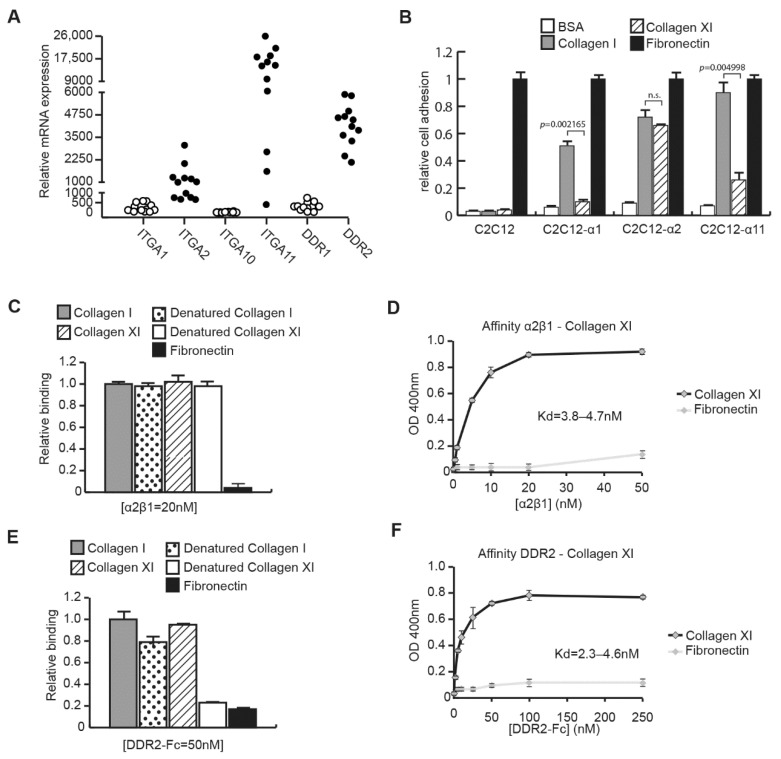
Integrin α2β1 and DDR2 bind to collagen type XI. (**A**) Expression of collagen receptors on CAF. Expression of integrin α1 (ITGA1), integrin α2 (ITGA2), integrin α10 (ITGA10), integrin α11 (ITGA11), DDR1 and DDR2 was analyzed by qPCR in CAF extracted from NSCLC patient tissue (*n* = 12). (**B**) Integrin binding to collagen type XI. C2C12 cells or C2C12 cells expressing either integrin α1 (C2C12-α1), integrin α2 (C2C12-α2) or integrin α11 (C2C12-α11) were seeded on fibrillar collagen type I and collagen type XI for 40 min. Cells were stained with crystal violet and absorbance was read at 595 nm. BSA and fibronectin were used as negative and positive controls for cell adhesion, respectively. Statistics were performed using Mann–Whitney test (*n* = 6, mean ± SD). (**C**) Direct interaction of integrin α2β1 and collagen type XI was confirmed by solid phase assay. Recombinant integrin α2β1 at the concentration of 20 nM was incubated for 3 h on immobilized collagen type I or collagen type XI coating or on heat-denatured collagens as the control. Fibronectin coating was used as the negative control. (**D**) Solid phase binding assay was carried out on immobilized collagen type XI or fibronectin using a different concentration of recombinant integrin α2β1. The dissociation constant Kd has been determined by the Scatchard equation. (**E**) Direct interaction of DDR2 and collagen type XI was confirmed by solid phase assay. Recombinant DDR2-Fc at the concentration of 50 nM was incubated for 3 h on immobilized collagen type I or collagen type XI coating or on-heat denatured collagens as the control. Fibronectin coating was used as the negative control. (**F**) Solid phase binding assay was carried out on immobilized collagen type XI or fibronectin using a different concentration of recombinant DDR2-Fc. The dissociation constant Kd has been determined by the Scatchard equation. n.s.; not significant.

**Figure 3 ijms-23-11722-f003:**
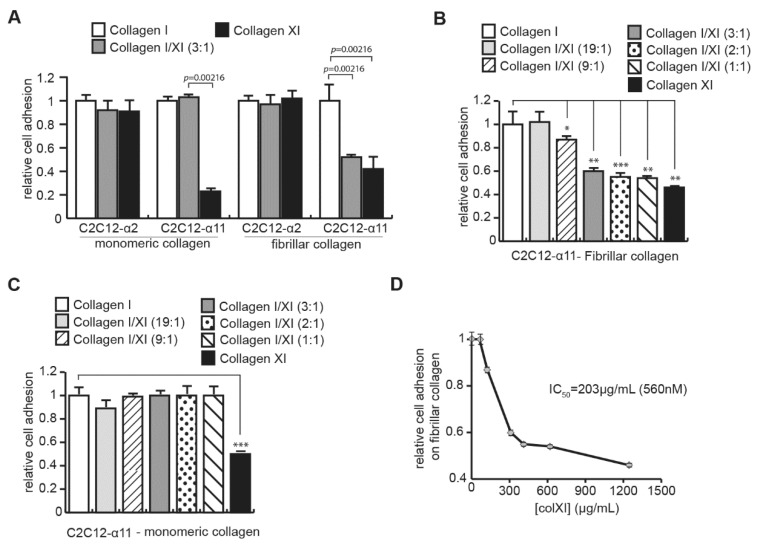
Collagen type XI restrains integrin-binding site availability on collagen type I. (**A**) Collagen type XI inhibits C2C12-α11 binding on fibrillar collagen type I. C2C12 cells expressing either integrin α2 (C2C12-α2) or integrin α11 (C2C12-α11) were seeded on monomeric or fibrillar collagen type I, collagen type XI or a mix of both collagens at a ratio of 3:1 for 40 min. Cells were stained with crystal violet and absorbance was read at 595 nm. Statistics were performed using Mann–Whitney test (mean ± SD). Dose-effect of collagen type XI on inhibition of cell attachment to collagen type I has been analyzed on fibrillar (**B**) and monomeric (**C**) collagens. C2C12-α11 cells were seeded on collagen type I, collagen type XI or a mix of both collagens at different ratios (total collagen concentration is constant) for 40 min. Cells were stained with crystal violet and absorbance was read at 595 nm. Statistics were performed using Mann–Whitney test (*, *p* = 0.00433; **, *p* = 0.00216; ***, *p* = 0.00499, mean ± SD). (**D**) Determination of the half-maximal inhibitory concentration (*IC_50_*) of collagen type XI to inhibit cell binding to fibrillar collagen type I. *IC_50_* has been calculated based on the results presented in the (**B**).

**Figure 4 ijms-23-11722-f004:**
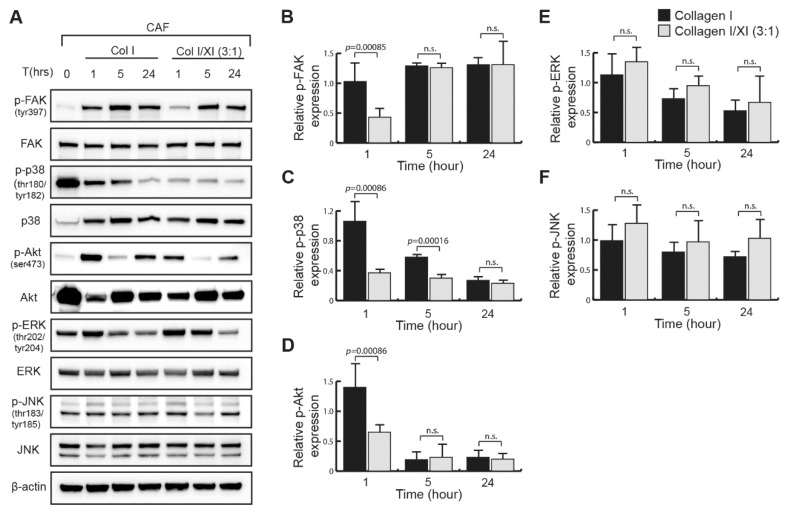
Collagen type I/XI matrices inhibit early FAK, p38 and Akt activation in CAFs. (**A**) Western blotting study of proteins extracted from CAFs seeded on a matrix of fibrillar collagen type I or a mix of both fibrillar collagen type I and XI at a ratio of 3:1 at different time points. CAFs were starved overnight before their interaction with collagens. Phosphorylation of FAK, Akt and MAPKs has been analyzed. Phosphorylated FAK (**B**), phosphorylated p38 (**C**), phosphorylated Akt (**D**), phosphorylated ERK (**E**) and phosphorylated JNK (**F**) band intensity were quantified by densitometry analysis and normalized to their respective total protein and β-actin. Statistics were performed using Mann–Whitney test (mean ± SD). n.s.; not significant.

**Figure 5 ijms-23-11722-f005:**
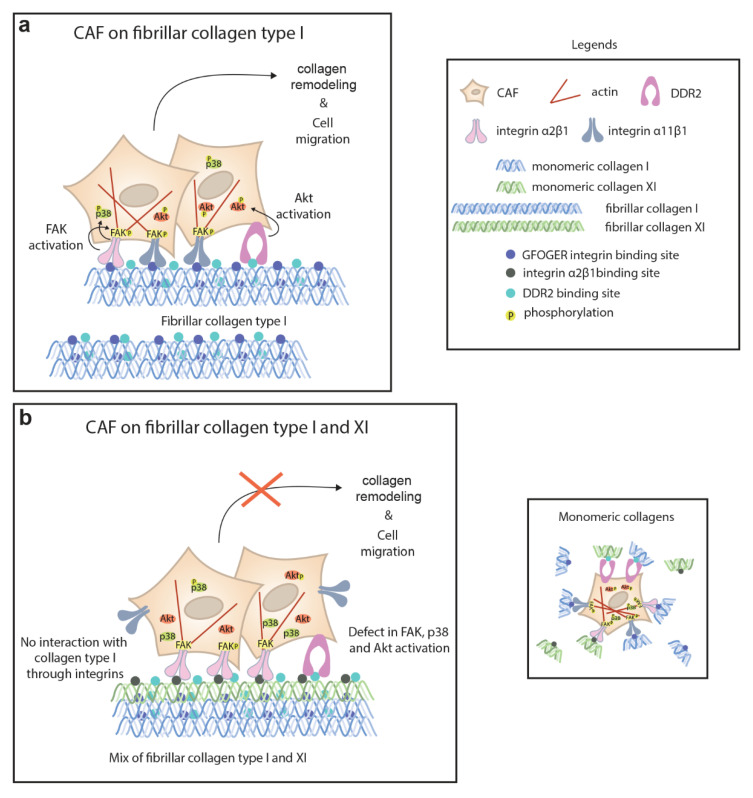
In collagen type I matrices, cancer-associated fibroblasts (CAFs) interact with fibrillar collagen through α2β1 and α11β1 integrins and DDR2 leading to the activation of FAK, p38 and Akt, which in turn promote collagen remodeling and cell migration. (**a**) Both α2β1 and α11β1 integrins recognize the GFOGER motif present in collagen type I. Integrins, via their link to actin cytoskeleton, control cell contractility and motility, whereas DDR2 could modulate integrin activity. In the presence of collagen type XI in collagen type I fibrils, integrin-binding sites on collagen type I are not available, inhibiting integrin α11β1 function. (**b**) Although integrin α2β1 and DDR2 can interact with collagen type XI, there is a defect of FAK and Akt long-term activation, resulting in inhibition of collagen remodeling and cell migration. On monomeric collagen, single molecules of collagen do not interfere between each other.

**Table 1 ijms-23-11722-t001:** Patient demographics, tumor stage and pathological diagnosis for tumors from which CAFs used in this study were isolated.

	Smoking History	Sex	Clinical Stage	Histology
CAF 094	Unknown	F	3A	ADC
CAF 448	Ex-Smoker	M	1B	ADC
CAF 453	Smoker	M	1A	ADC
CAF 455	Never	F	2A	ADC
CAF 458	Never	M	3	SqCC
CAF 462	Smoker	F	1B	SqCC
CAF 466	Ex-Smoker	M	1B	ADC
CAF 472	Never	F	1A	ADC
CAF 474	Unknown	F	1B	ADC
CAF 476	Ex-Smoker	F	3A	ADC
CAF 480	Ex-Smoker	M	1B	ADC
CAF 481	Never	F	1B	ADC
CAF 482	Ex-Smoker	M	2B	SqCC
CAF 487	Ex-Smoker	M	3A	SqCC

ADC—adenocarcinoma; SqCC—squamous cell carcinoma.

**Table 2 ijms-23-11722-t002:** Concentration of collagens in fibrillar collagen coating. The indicated ratios are for collagen type I: collagen type XI. The final collagen concentration is 1.24 mg/mL.

	(Collagen Type I) mg/mL	(Collagen Type XI) mg/mL	(Collagen Type I) µM	(Collagen Type XI) µM
Coll I	1.24	0	4.1	0
Coll I/XI (19:1)	1.178	0.062	3.93	0.17
Coll I/XI (9:1)	1.116	0.124	3.72	0.34
Coll I/XI (3:1)	0.93	0.31	3.1	0.85
Coll I/XI (2:1)	0.83	0.41	2.75	1.14
Coll I/XI (1:1)	0.62	0.62	2.07	1.7
Coll XI	0	1.24	0	3.4

**Table 3 ijms-23-11722-t003:** RT-qPCR primer sequences.

Gene	Primer Sequence
*ITGA1*	Forward 5’-CACTGTTGTTCTACGCTGCT-3’ Reverse 5’-ACGGAGAACCAATAAGCAC-3’
*ITGA2*	Forward 5’-CTAGAAGCCCATCCTGTGCC-3’ Reverse 5’-GGTTACTGCCAGTCAGCCAA-3’
*ITGA10*	Forward 5’-CATCACCCACGCCTATTCCC-3’ Reverse 5’-TGCCTCCCCTTTGTTAGCAC-3’
*ITGA11*	Forward 5’-GCCTCCAGTATTTTGGCTGC-3’ Reverse 5’-GCTCAAAGTGGAGGCTGGC-3’
*DDR1*	Forward 5’-GATCTCGACTCCGCTTCAAG-3’ Reverse 5’-CAAAGGGTGTCCCTTACGC-3’
*DDR2*	Forward 5’-AACGAGAGTGCCACCAAT-3’ Reverse 5’-ACTCACTGGCTTCAGAGCG-3’
*RPS13*	Forward 5’-GTTGCTGTTCGAAAGCATCTTG-3’ Reverse 5’-AATATCGAGCCAAACGGTGAA-3’

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
