# Peer review of "Collagen Type XI Inhibits Lung Cancer-Associated Fibroblast Functions and Restrains the Integrin Binding Site Availability on Collagen Type I Matrix"

_ijms, 2022, doi:10.3390/ijms231911722_

Round 1

Reviewer 1 Report

The manuscript “Collagen type XI inhibits lung cancer-associated fibroblast functions and restraint the integrin binding site availability on collagen type I matrix” by Cédric Zeltz et al. analyse how collagen type XI in cell culture matrix could influence lung primary CAFs, from NSCLC resected tumours, features. Cancer-associated fibroblasts (CAFs) are one of the major cellular components of the tumour microenvironment. Therefore, CAFs play active roles in the tumour microenvironment and tumourigenesis, influencing tumour progression and metastasis. Specifically, the authors focus on the role of collagen XI in NSCLC stroma and unexpectedly, they find that collagen XI inhibits CAFs-mediated collagen remodelling and cell migration in collagen I matrices. To explore the mechanism of collagen type XI inhibitory effect on CAFs the authors describe how NSCLC-CAFs interact with collagen type XI through integrin α2β1 and DDR2 and a defect in FAK, p38 and Akt signalling pathway in presence of collagen type XI

In my opinion, the following points should be changed before accepting the manuscript: 

1I have detected the spelling error ‘red’ in lines 73, 190 or 194 and also, I would recommend editing of English language and style. All the paper is written in past tense and in addition to that, sometimes it is difficult to understand which is the hypothesis or question to be tested and which is the final experimental result. 

On the other hand, the figures have excellent quality and a nice presentation which have helped the paper comprehension.

2- In figure 1 and figure 4 there is no information about the CAFs that have been tested for the experiments. Have all the experiments been done with just one, several or all of the CAFs described in line 328? I would consider showing all the points, as in figure 2A. 

I am curious about this point as I have observed that the tumours from which CAFs were isolated are quite diverse from a pathological diagnosis point of view (adenocarcinoma and squamous cell carcinoma) and the tumour stage of these neoplasias (from stage 1A to stage 3A). As far as I know, the tumour microenvironment is quite affected by the type of neoplasia (ADC and SqCC) and tumour stage. Could these differences be affecting the endogenous properties and behaviour of CAFs? and therefore be an important point for figure 1, supplementary figure 3 and figure 4 results?

3Concerning figure 4 I have also a problem with the data represented in panel D -Relative pAKT expression- and the Western Blot result of pAKT in panel A. I only detect differences at 5 hours and not at 24 hours. Representing all the points, as in figure 2A, could mitigate this problem. 

4- Sure I am missing something, but I fail to understand how or why C2C12-α11 having a defect in binding collagen XI (Fig2.B and Supp. Fig2.B) can induce collagen matrix remodelling even better than C2C12-α2 (Supp. Fig2.B. Collagen I/XI (3:1) at 5h). 

[Also in this figure there is an error in the pattern of C2C12-αCollagen I/XI (3:1) at 24h]

By the results presented in the paper, I interpret that matrix remodelling has an important impact on CAFs migration. Have you checked the C2C12-α2 and C2C12-α11 migration in collagen I, collagen XI and Collagen I/XI (3:1)?

5In line 111 complete the sentence: 'suggesting that collagen type XI specifically inhibited CAFs migration'

6In line 126: ‘indicating that integrin α2β1 is the only collagen-binding integrin on CAFs..’ Only is quite categorical I would change to main.

7- From line 201 to 202: α11β1 recognize the same sequence on collagen type I [24], we concluded that collagen type XI also inhibits the interaction between integrin α2β1 and collagen type I. There are no experimental results to support this conclusion.

Author Response

Reviewer#1

We thank the reviewer for his useful comments. Below we discuss the comments and the changes done in the manuscript.

The manuscript “Collagen type XI inhibits lung cancer-associated fibroblast functions and restraint the integrin binding site availability on collagen type I matrix” by Cédric Zeltz et al. analyse how collagen type XI in cell culture matrix could influence lung primary CAFs, from NSCLC resected tumours, features. Cancer-associated fibroblasts (CAFs) are one of the major cellular components of the tumour microenvironment. Therefore, CAFs play active roles in the tumour microenvironment and tumourigenesis, influencing tumour progression and metastasis. Specifically, the authors focus on the role of collagen XI in NSCLC stroma and unexpectedly, they find that collagen XI inhibits CAFs-mediated collagen remodelling and cell migration in collagen I matrices. To explore the mechanism of collagen type XI inhibitory effect on CAFs the authors describe how NSCLC-CAFs interact with collagen type XI through integrin α1 and DDR2 and a defect in FAK, p38 and Akt signalling pathway in presence of collagen type XI

In my opinion, the following points should be changed before accepting the manuscript: 

1- I have detected the spelling error ‘red’ in lines 73, 190 or 194 and also, I would recommend editing of English language and style. All the paper is written in past tense and in addition to that, sometimes it is difficult to understand which is the hypothesis or question to be tested and which is the final experimental result. 

On the other hand, the figures have excellent quality and a nice presentation which have helped the paper comprehension.

Response: We apologize for the spelling errors, the sentence has been copy-paste in the figure legends, that is why it has been repeated, but we should have notice it. we have now corrected these mistakes.

2- In figure 1 and figure 4 there is no information about the CAFs that have been tested for the experiments. Have all the experiments been done with just one, several or all of the CAFs described in line 328? I would consider showing all the points, as in figure 2A. 

Response: We did not use the CAFs we mentioned in all experiments. It has been set up as follow:

Adhesion (Fig1A): CAF094, CAF474, CAF487

Collagen contraction (Fig1B): CAF094, CAF474, CAF487, CAF481

Migration (Fig1C-E): CAF481, CAF487

Western Blot (Fig4): CAF481, CAF487

qPCR (Fig2A): CAF448, CAF453, CAF455, CAF458, CAF462, CAF466, CAF472, CAF474, CAF476, CAF480, CAF481, CAF482

I am curious about this point as I have observed that the tumours from which CAFs were isolated are quite diverse from a pathological diagnosis point of view (adenocarcinoma and squamous cell carcinoma) and the tumour stage of these neoplasias (from stage 1A to stage 3A). As far as I know, the tumour microenvironment is quite affected by the type of neoplasia (ADC and SqCC) and tumour stage. Could these differences be affecting the endogenous properties and behaviour of CAFs? and therefore be an important point for figure 1, supplementary figure 3 and figure 4 results?

Response: This is an interesting point. These CAFs have been isolated and characterized in our previous publication (Hao et al., Neoplasia 2019, 21, 482–493, doi:10.1016/j.neo.2019.03.009), in which we have subclassed the CAFs into two populations based on desmoplasia. High desmoplastic CAFs (e.g. CAF487, sqCC, stage 3A) display higher ability to promote tumorigenicity compared to low desmoplastic CAFs (e.g. CAF481, ADC, stage 1B). However, in the present study, in regard to effect of the presence of collagen type XI in matrices, CAFs (including 481 and 487) were similarly affected, independently of the histology or the stage of the tumor from which they were isolated.

3- Concerning figure 4 I have also a problem with the data represented in panel D -Relative pAKT expression- and the Western Blot result of pAKT in panel A. I only detect differences at 5 hours and not at 24 hours. Representing all the points, as in figure 2A, could mitigate this problem. 

Response: We thank the reviewer for his concern. Actually, for pAkt, in figure 4, we only detect differences between collagen I and collagen I/XI (3:1) at 1 hour. There were no differences at 5 and 24 hours. However, when CAFs are seeded on collagen XI only, we observed a difference in Akt phosphorylation at 24 hours only (supplementary figure S3).

4- By the results presented in the paper, I interpret that matrix remodelling has an important impact on CAFs migration. Have you checked the C2C12-α2 and C2C12-α11 migration in collagen I, collagen XI and Collagen I/XI (3:1)?

Response: The ability of reorganizing the matrix is an important feature of some CAF subpopulations in tumors. By remodeling the matrix, CAFs can align collagen and fibronectin matrices, creating "highways" for CAF migration and tumor invasion. The effect of the presence of collagen type XI on cell migration seems to be cell type-dependent: It inhibits lung CAF migration, but acts differently on lung tumor cells (an aspect we noticed on a parallel study without the presence of CAFs). We have just used mouse C2C12 cell lines as a tool to study the interaction between integrins and collagen matrices in a simpler way (CAFs expressing several collagen binding integrins). Since C2C12 expressing collagen binding integrins have the ability to contract collagen type I matrix (due to their myoblastic origin), we used them in this assay as control, but with unexpected results (collagen type XI affected C2C12-a11 mediated collagen contraction despite the low binding of a11b1 to this collagen). However, we are not interested on investigating the impact of collagen type XI on migration of the C2C12 cell lines, since it is not relevant for this study. C2C12 cells are immortalized myoblasts and differ from primary CAFs, regarding their biological functions.

Sure I am missing something, but I fail to understand how or why C2C12-α11 having a defect in binding collagen XI (Fig2.B and Supp. Fig2.B) can induce collagen matrix remodelling even better than C2C12-α2 (Supp. Fig2.B. Collagen I/XI (3:1) at 5h). 

Response: We understand the confusion of the reviewer. The C2C12-a11 cells we used had a higher ability to contract collagen type I matrices than C2C12-a2 cells. This can be due for example to higher contractility features or to higher secretion of other extracellular matrix protein, like fibronectin, known to bind to collagen type I and participate to collagen contraction through other integrins. Thus, it is difficult to directly compare between cell types. Before we performed this experiment, we hypothesized that the inhibition observed in CAFs was only due to direct interaction between integrin a2b1 and collagen type XI.  In this context, since C2C12-a11 cells bind poorly to collagen type XI, we expected no differences in C2C12-a11-mediated collagen contraction between collagen I and collagen I/XI (3:1) conditions. However, we observed an inhibition that prompted us to investigate whether collagen type XI can inhibit the interaction between integrin a11b1 and collagen type I. The defect of C2C12-a11 in binding collagen XI is still represented by its fail to contract the collagen type XI matrix at 24 hours (in contrast to C2C12-a2). In collagen I/XI (3:1) matrix, the interaction of C2C12-a11 with collagen I is only partially inhibited, and with the reasons we have exposed in the beginning of this response, this could explain the differences between C2C12-a11 and C2C12-a2 cells.

[Also in this figure there is an error in the pattern of C2C12-α2 Collagen I/XI (3:1) at 24h]

Response: We apologize for this mistake. We have now corrected it with the good pattern.

5- In line 111 complete the sentence: 'suggesting that collagen type XI specifically inhibited CAFs migration'

Response: We agree and have now completed the sentence.

6- In line 126: ‘indicating that integrin α1 is the only collagen-binding integrin on CAFs..’ Only is quite categorical I would change to main.

Response: We have followed the reviewer's recommendation and changed "only" to "main".

7- From line 201 to 202: α11β1 recognize the same sequence on collagen type I [24], we concluded that collagen type XI also inhibits the interaction between integrin α1 and collagen type I. There are no experimental results to support this conclusion.

Response: We agree with the reviewer's comment and we have modified the sentence to now read " ...α11β1 recognize the same sequence on collagen type I, we suggest that collagen type XI may also inhibit the interaction between integrin α2β1 and collagen type I."

Reviewer 2 Report

Thank you for the chance you gave me to read this interesting study entitled “Collagen type XI inhibits lung cancer-associated fibroblast functions and restraint the integrin binding site availability on collagen type I matrix” by Zeltz et al. In this original research paper, the authors evaluated the role of collagen type XI in NSCLC stroma showing that collagen type XI inhibits CAF-mediated collagen remodeling and cell migration. This is a very interesting study presenting very new data regarding the interplay of collagen type XI with lung cancer stroma. This topic has great importance, the manuscript is well-written and conclusions are based on the findings. I think that this study in the current form satisfies the appropriate criteria for publication in this journal, however, some minor points need to be treated before publication.

 Minor points:

High similarity rate (26%) based on the Turnitin. Please, rephrase some sentences.

Please, add a paragraph describing the limitations of the study.

All abbreviations should be expanded at their first mention.

Lines 265-267: Please, rephrase.

Author Response

Reviewer#2

We thank the reviewer for his useful comments. Below we discuss the comments and the changes done in the manuscript.

Thank you for the chance you gave me to read this interesting study entitled “Collagen type XI inhibits lung cancer-associated fibroblast functions and restraint the integrin binding site availability on collagen type I matrix” by Zeltz et al. In this original research paper, the authors evaluated the role of collagen type XI in NSCLC stroma showing that collagen type XI inhibits CAF-mediated collagen remodeling and cell migration. This is a very interesting study presenting very new data regarding the interplay of collagen type XI with lung cancer stroma. This topic has great importance, the manuscript is well-written and conclusions are based on the findings. I think that this study in the current form satisfies the appropriate criteria for publication in this journal, however, some minor points need to be treated before publication.

 Minor points:

High similarity rate (26%) based on the Turnitin. Please, rephrase some sentences.

Response: We thank the reviewer for his concern. We have run a plagiarism checker (Ouriginal by Turnitin). We found only 13% similarities, most of them corresponding to the list of references and few minor in Materials and Methods section only. We have included the report for the review process only.

Please, add a paragraph describing the limitations of the study.

Response: We have added a paragraph at the end of the discussion section related to the limitations of our study:

"We have shown that collagen type XI restrained interaction of integrin a11b1 with collagen type I, but we are lacking direct evidence in regard to integrin a2b1. We did not find an appropriate method to test this hypothesis for integrin a2b1, due to: 1) integrin a2b1 binds to both collagens and 2) this inhibition occurs only when collagens form fibrils. Instead, we speculated based on the sequence recognition by both integrins on collagen type I and we assumed that since they are known to both interact with the same collagen type I sequence, in this context if integrin a11b1-collagen type I interaction is inhibited, this should also be true for integrin a2b1. Another limitation is the use of bovine collagens instead of human collagens due to technical limitation. Although collagens are highly conserved between the two species (identity >95% between the human and bovine protein sequence of both COL1A1 and COL11A1), the interaction between bovine collagens and human integrins may differ from the in vivo affinities in human tumors, and small variations could thus exist."

All abbreviations should be expanded at their first mention.

Response: We have now expanded abbreviations at their first mention in the text.

Lines 265-267: Please, rephrase.

Response: We have modified the sentence to now read "Therefore, we suggest that the sub-populations of CAF expressing collagen type XI may exhibit anti-tumoral activities rather than promoting tumorigenicity."

Reviewer 3 Report

1.      In Figure 1a, it is necessary to indicate the number of measurements and what values are shown on the graph (mean±SD?)

2.      It is necessary to indicate which line of cancer cells was used when calculating migrated area in collagen matrices for heterospheroids (Figure 1d-e), also it is necessary to indicate the number of measurements for this quantitation.

3.      I did not find information about which reference gene was used to determine the relative expression of mRNA during RT-PCR and how this value was calculated (Figure 2a)

4.      Figures 2b and 3 show the parameter «relative cell adhesion». It is clear from the text which method was used to determine this parameter, but it is not clear how it was calculated (what is relative cell adhesion=1?).

5.      In Figures 3a and 3c there are columns corresponding to the relative adhesion of C2C12-α11 cells to monomeric collagen. Why are these values different in the two graphs?

Round 2

Reviewer 1 Report

All the questions have been answered thus the paper, in the current
form, satisfies the criteria for publication.

Reviewer 3 Report

My comments are taken into account, the article can be published in the current form.